# Effects of alcohol consumption on the prevalence and incidence of non-alcoholic fatty liver disease: A systematic review and meta-analysis

Jingkai Yuan[1], Zhiheng Chen[2], Yukai Gu[3], Yaobin Liang[1], Zhiqi Yao[4]*

1 The Second Clinical Medical College of Guangzhou University of Chinese Medicine, Guangzhou, China,
2 The Affiliated TCM Hospital of Guangzhou Medical University, Guangzhou, China, 3 The First Clinical
Medica lCollege of Guangzhou University of Chinese Medicine, Guangzhou, China, 4 The Affiliated
Shunde Hospital of Jinan University, Foshan, China

* yaozhiqi@sddermyy.com

## Abstract

### Background and aims

Nonalcoholic fatty liver disease (NAFLD) is one of the most prevalent diseases worldwide, with its prevalence and incidence continually increasing. However, the impact of alcohol consumption on the development and progression of hepatic steatosis has not been systematically investigated. Our aim was to estimate the impact of alcohol consumption on the development and progression of NAFLD.

### Methods

The Preferred Reporting Items for Systematic Reviews and Meta-Analyses (PRISMA) standards were followed. We conducted a search of Web of Science, PubMed, Embase, and the Cochrane Library without language restrictions, covering the period from inception to December 31, 2023. Abstract screening, full-text review, and data extraction were performed in duplicate.

### Results

We identified 16 articles that reported adjusted data (Japan = 7, other countries = 9). Random-effects categorical meta-analyses were conducted to compare alcohol consumption levels (< 20 g/day for women and < 30 g/day for men) with those of non-drinkers. A total of 299,955 participants were included, with 63,693 cases of NAFLD. Overall, there was no significant difference in the prevalence of NAFLD between non-drinkers and light drinkers (RR = 0.99, 95% CI, 0.85–1.15). In subgroup analyses, no differences were observed between the Japan cohort (RR = 1.01, 95% CI, 0.81–1.25) and the participants from other countries (RR = 0.96, 95%

**Data availability statement:** The data underlying the results presented in the study are available from https://pubmed.ncbi.nlm.nih.gov/.

**Funding:** The author(s) received no specific funding for this work.

**Competing interests:** The authors have declared that no competing interests exist.

CI, 0.76–1.21). Gender-specific subgroup analyses indicated that light drinking was associated with a reduced prevalence of NAFLD in men (RR = 0.82, 95% CI, 0.79–0.85), while no significant association was found in women (RR = 0.90, 95% CI, 0.60–1.36). Regarding incidence, non-drinkers were consistently associated with a substantially lower incidence of NAFLD compared to light drinkers (RR = 1.18, 95% CI, 1.08–1.30).

## Conclusion

This study summarizes the impact of alcohol consumption on the incidence and prevalence of NAFLD. In prevalence, light drinking in men was associated with a lower prevalence of NAFLD, whereas light drinking was associated with a higher incidence of NAFLD.

## Introduction

Non-alcoholic fatty liver disease (NAFLD) has a global prevalence of 25% and is a leading cause of cirrhosis and hepatocellular carcinoma. NAFLD encompasses a disease continuum that ranges from steatosis, with or without mild inflammation (non-alcoholic fatty liver), to non-alcoholic steatohepatitis (NASH), which is characterized by necroinflammation and faster fibrosis progression compared to non-alcoholic fatty liver [1]. Moreover, the prevalence of NAFLD continues to rise worldwide [2]. It is projected that by 2030, NAFLD will become the most common indication for liver transplantation in Western countries [3].

The diagnosis of NAFLD requires the exclusion of both secondary causes and a daily alcohol consumption of ⩾30 g for men and ⩾20 g for women [4]. While early epidemiological studies indicated that moderate alcohol consumption does not contribute to the development of NAFLD [5], increasing evidence suggests that alcohol consumption can be the risk factor for NAFLD [6]. However, current guidelines do not provide specific recommendations regarding alcohol consumption for patients with NAFLD. The 2023 American Association for the Study of Liver Diseases NAFLD guidance states that only patients with clinically significant hepatic fibrosis (≥F2) should be advised to abstain from alcohol use [7]. In contrast, European clinical guidelines suggest abstinence or strict adherence to alcohol intake below the risk threshold (30 g in men and 20 g in women) in individuals with NAFLD and the absence of cirrhosis [4]. Likewise, the Latin American Association for the Study of the Liver NAFLD practice guidance points out that patients with NAFLD are advised to avoid alcohol. However, if patients choose to drink, the recommendation is that men should limit their intake to less than 21 drinks per week, while women should consume no more than 14 drinks per week [8].

Data indicate that heavy alcohol consumption is often underestimated among patients diagnosed with NAFLD. Furthermore, the interaction between alcohol and metabolic factors can exacerbate the progression of liver disease [9]. This raises an important question: What is the impact of light drinking on the prevalence and

incidence of NAFLD? Previous studies examining the effects of light alcohol consumption on NAFLD prevalence and incidence have suggested that light drinking can reduce both [5]. However, these studies defined light drinking as less than 40 g/day for men [10] and 20 g/day for women [11]. Since the last review on the impact of light drinking on NAFLD prevalence and incidence, numerous new studies have emerged, increasingly indicating that light alcohol consumption does not reduce the prevalence or incidence of NAFLD.

Given the above, this study aims to define light drinking as < 20 g/day for women and < 30 g/day for men [4], and it will compare the effects of light drinking with those of non-drinking on the prevalence and incidence of NAFLD. Ultimately, this research will enhance our understanding of the impact of alcohol consumption on NAFLD and guide people in recognizing the associated risks of drinking while managing their alcohol intake.

## Methods

A systematic review and meta-analysis were conducted according to a predetermined protocol, submitted to and registered with PROSPERO (CRD42024497483). The review process followed the Meta-analysis of Observational Studies in Epidemiology (MOOSE) guidelines.

### Search strategy and selection criteria

We searched in Web of Science, PubMed, Embase, and the Cochrane Library without language restrictions from inception to December 31, 2023. A complete list of search terms is provided in the S1 Appendix in S1 File. After excluding duplicates, citations were independently screened by two authors (YJK, CZH). The full texts of the selected studies were obtained and examined independently in duplicate (by YJK and CZH) to determine their eligibility for inclusion in the systematic review. In cases of discrepancies,consensus was reached after discussion with the senior author (YZQ).

A search of the reference lists from previous systematic reviews on prevalence [10,11] and incidence [11] was also conducted, applying the same inclusion and exclusion criteria used in our search to identify any publications that had not been previously captured.

We established criteria to select research publications that provided accurate estimates for the general adult population across different world regions. We included original descriptive research publications, including cross-sectional studies, cohort studies, and prospective studies. These studies were required to report crude data necessary to calculate the prevalence or incidence rate estimates for non-drinkers and light drinkers, including study date, sample size, number of NAFLD cases, diagnostic methods, alcohol consumption assessment methods, the year of study (mid-point of the study period for incidence studies), and the mean or median duration of follow-up. Studies that did not meet the criteria for adult populations were excluded. Furthermore, we excluded studies that did not specify amounts of alcohol consumption classifications, those with classifications of alcohol consumption that did not align with the defined boundaries for non-drinkers and light drinkers, studies that did not provide specific case numbers, and those that failed to exclude other causes of liver disease (such as viral hepatitis, alcoholic liver disease, etc.). We did not search for unpublished reports or grey literature. In addition, reviews, pathology reports, non-human studies, conference abstracts, letters, editorials, and rejected manuscripts were also excluded.

### Data extraction from selected studies

Data extraction was performed using a standardized data collection table by two authors (YJK, GYK). The third author (LYB) reviewed the table for accuracy. The variables included were the first author, year of publication, country, research type, follow-up time, alcohol consumption classification, total number of people, age, gender, number of cases, diagnostic method, and method of alcohol assessment. We only included unique, non-overlapping samples to avoid multiple publication biases. After a cross-check process, a consolidated entry form was subsequently developed.

 

## Method of quality appraisal

The evaluation criteria for an observational study of the Agency for Healthcare Research and Quality (AHRQ) were adapted to evaluate the quality of cross-sectional studies [12]. In contrast, the Newcastle–Ottawa Scale (NOS) was used to assess the quality of case-control studies [13]. The AHRQ evaluation criteria for an observational study consisted of 11 items, and each item of the AHRQ criteria was answered as "yes," "no," or "not reported." Items with a "yes" scored one point, while items with "no" and "not reported" scored zero. Scores between 8 and 11 were regarded as high quality, 4–6 as moderate quality, and 0–3 as low quality. The NOS assigns a maximum of nine points to eight items in three categories: selection, comparability, and outcome. A study can be awarded a maximum of one point for each numbered item within the selection and outcome categories and a maximum of two points for items in the comparability category. A study was regarded as high quality if it scored more than six. Two researchers, YJK and CZH, conducted the risk of bias and literature quality assessment, with discrepancies resolved by the consensus of a third author, YZQ.

## Data synthesis and analyses

In instances where means and standard deviations were not reported, we estimated these values from medians and percentiles [14]. Multiple means and standard deviations were combined using the StatsToDo online web program (https://www.statstodo.com/index.php).

In the meta-analysis, the prevalence and incidence of NAFLD among non-drinkers and light drinkers were calculated based on the number of reported original cases, sample size, and follow-up duration. Publication bias was assessed by using the Egger test and by constructing the funnel plot. A random-effects model was utilized to compare the prevalence of NAFLD in non-drinkers and light drinkers. We estimated heterogeneity between studies using Cochran's Q statistic ($p < 0.05$ indicates moderate heterogeneity) and the $I^2$ statistic ($\geq 50\%$ indicates moderate heterogeneity). Subgroup analyses were conducted to investigate sources of heterogeneity, testing individual associations between the pooled estimates and the following covariates: sex and country or region.

Using the random-effects model, we determined the pooled prevalence of NAFLD in non-drinkers and light drinkers. The random-effects model was also used to pool the incidence rates of NAFLD among non-drinkers and light drinkers without NAFLD at baseline. All meta-analytic analyses were performed using the meta-packages in R statistical software (version 4.3.1).

# Results

The literature search returned 10,455 publications from various databases: PubMed (1,506), EMBASE (5,667), Web of Science (3,160), and Cochrane Library (122). An additional 16 studies were included from other sources (reference lists of relevant publications). After removing duplicates, 6,001 studies remained for further evaluation. Following a screening of titles and abstracts, 5,914 studies were excluded, resulting in a total of 87 full texts to be assessed. Upon reviewing these full texts, 71 studies were subsequently excluded. Ultimately, 16 studies were retained for review. Among these, 14 studies were eligible for prevalence analysis, and four were suitable for incidence analysis (Fig 1).

The characteristics of the included studies for the analyses of NAFLD prevalence and incidence are presented in Table 1. We specifically included studies that met the criteria for light alcohol consumption and non-alcohol consumption, excluding any that did not meet these criteria. Most of studies originated from mainland Japan (7 studies), followed by China and Germany, each contributing two studies. The 14 studies included in the NAFLD prevalence analysis comprised a sample population of 81,069 individuals. In contrast, four studies included in the NAFLD incidence analysis encompassed a sample population of 218,886 individuals.

## Prevalence

Among the 14 included studies, there were 37,633 non-drinkers and 43,436 light drinkers. The prevalence of NAFLD among non-drinkers was found to be 33% (95% CI, 0.22–0.46), while the prevalence among light drinkers was 31% (95%

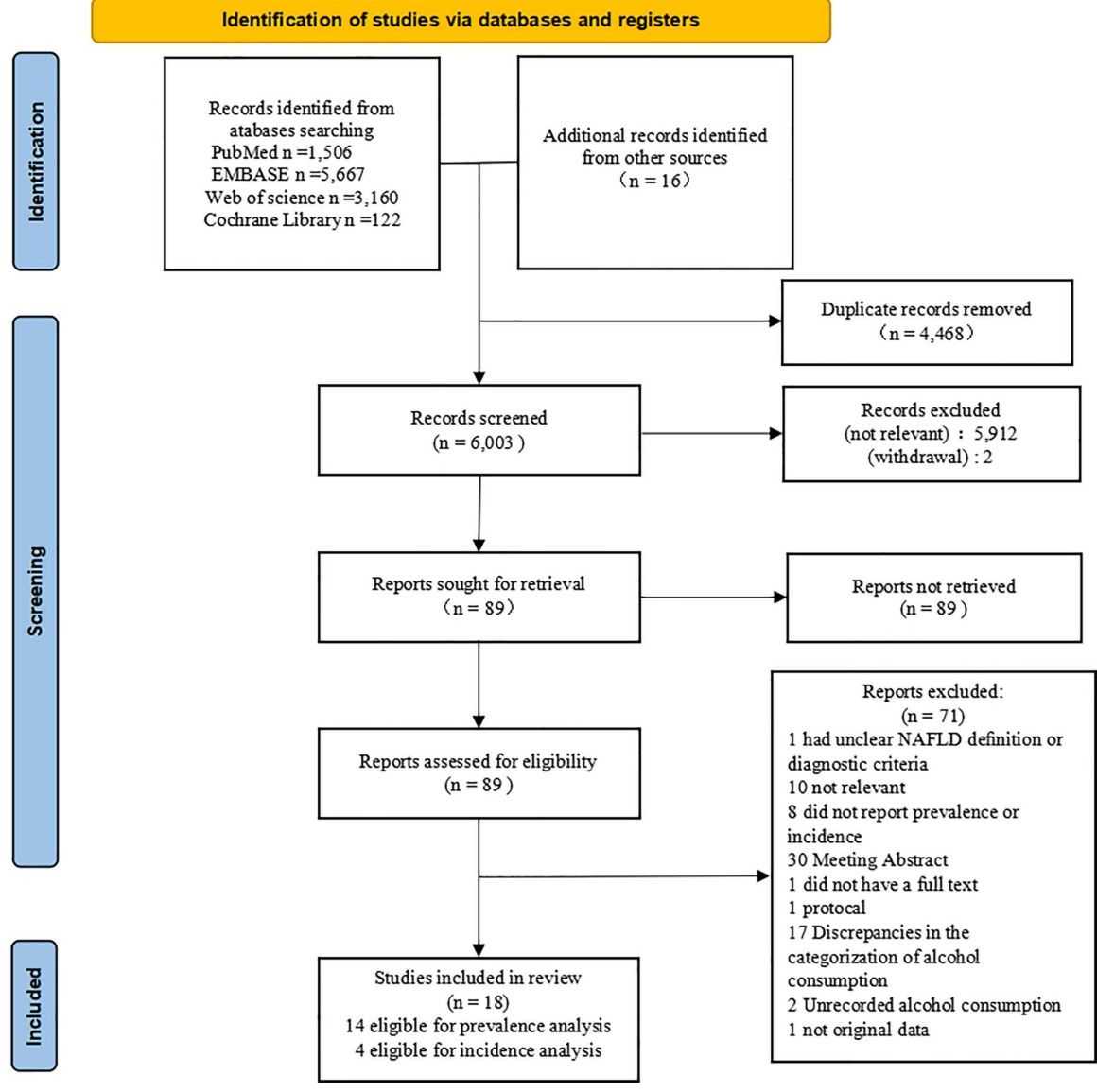

**Fig 1. Study selection.**

CI, 0.22–0.43) (Fig 2). When these two groups combined, the overall prevalence of NAFLD was calculated to be 32% (95%CI, 0.25–0.40).

Fig 3 displays the relative risk (RR) of the relationship between light drinkers and NAFLD compared to non-drinkers. Using a random-effects model, no significant difference in prevalence was observed between two groups (RR = 0.99, 95% CI, 0.85–1.15).

We conducted a subgroup analysis by categorizing all studies into two groups: one for Japan and another for other countries. In the Japan group, the prevalence of NAFLD was 25% (95% CI, 0.17–0.35), whereas in the other countries group, the prevalence of NAFLD was 43% (95% CI, 0.23–0.66). The relative risk (RR) of the relationship between the prevalence of light drinkers and non-drinkers was assessed within the Japan group. Using a random-effects model, no significant difference in prevalence was observed between the two groups (RR = 1.01, 95% CI, 0.81–1.25). In the group

**Table 1. Overall characteristics of retained studies.**

| Author | Year | Country or region | Study design | Follow up time | Alcohol consumption measurement | Total number | Age(Mean, SD or IQR) | Men/Woman | Number of cases(total number) | Diagnostic method | Alcohol assessment method |
|---|---|---|---|---|---|---|---|---|---|---|---|
| Hamaguchi [15] | 2005 | Japan | prospective cohort study | NA | Non-drinker Light drinkers: <20g/day | 4401 | 47.6±8.8 | 2572/1829 | 812 N:275(1579):M/W:170(595)/105(981) L:537(2825):M/W:464(1977)/73(848) | Ultrasonography | Questionnaire |
| Yamada [16] | 2009 | Japan | cross-sectional and retrospective longitudinal study | 5Y | Non-drinker Occasional drinker: <1drink/day Moderate drinker:1 drink/day Heavy drinker: >1drink/day | Base: 63447 Last: 10424 | 48.6±11.8 50.1±10.1 | 32438/31009 5444/4980 | Base:13341 N:4255(25179) M/W:2006(7039)/2249(18140) O:6210(20918) M/W:3015(10964)/3195(9954) M:1844(11248) M/W:1674(8953)/170(2295) H:1032(5910) M/W:1022(5352)/10(558) Last:1178 N:426(4139) M/W:173(1057)/253(3082) O:386(3139) M/W:285(1704)/101(1489) M:292(2107)M/W:227(1758)/20(349) H:119(985)M/W:115(925)/4(60) | Ultrasonography | Questionnaire |
| Cotrim [17] | 2009 | Brazil | cross-sectional study | NA | G1:20-40g/day G2:<20g/day G3:Non-drinker | 132 | 37.3±11 | 41/91 | 118 G1:17(19) G2:50(56) G3:51(57) | Liver biopsy | Interview |
| Caballeria [18] | 2010 | Spain | multicentre, cross-sectional, populational study | NA | Non-drinker Drinker: <30g/day in men; <20g/day in women | 759 | 53±14 | 323/443 | 195 N:122(534) D:73(225) | Ultrasonography | Anamnesis |
| Moriya [19] | 2010 | Japan | cross-sectional study | NA | Non-drinker 0.1-139.9g/week >139.9g/week | 7112 | 48.3±9.4 | 4957/2155 | 1874 Man: N:507(1268) 0.1-139.9g:460(1394) >139.9g:596(2295) Woman: N:246(1535) 0.1-139.9g:52(516) >139.9g:13(104) | Ultrasonography | Questionnaire |
| 土居忠 [20] | 2010 | Japan | Cross-sectional study | NA | Non-drinker Minima: <20g/d Light:20-40g/d Moderat:40-60g/d Heavy:>60g/d | 3185 | 50.6±8 | 2246/939 | 1076 N:258(755) Min:378(1142) L:189(587) M:174(501) H:77(200) | Ultrasonography | Questionnaire |
| Hamabe [21] | 2011 | Japan | retrospective follow-up study | 10Y | Non-drinker Light drinkers: <20g/day | BASE: 2029 LAST: 1560 | 50.7±9.2 | 1114/915 772/788 | Base: 469 N:159(814) L:310(1215) Last:266 N:104(655) L:162(905) | Ultrasonography | Questionnaire |

*(Continued)*

PLOS One | https://doi.org/10.1371/journal.pone.0330105 September 19, 2025

Table 1. (Continued)

| Author | Year | Country or region | Study design | Follow up time | Alcohol consumption measurement | Total number | Age(Mean, SD or IQR) | Men/Woman | Number of cases(total number) | Diagnostic method | Alcohol assessment method |
|---|---|---|---|---|---|---|---|---|---|---|---|
| Wong [22] | 2012 | China | cross-sectional study | NA | Non-drinkers drinkers: <140g/week | 910 | 48±11 | NA | 252 N:188(719) <140g:64(191) | Proton-magnetic resonance spectroscopy | Questionnaire |
| Moriya [23] | 2013 | Japan | cross-sectional study | NA | Non-drinkers <70g/week, 70-<140g/week ≥140g/week | 4921 | 46.4±8.9 | W:4921 | 681 N:527(3403) <70g:82(864) 70-<140g:35(355) ≥140g:37(299) | Ultrasonography | Questionnaire |
| Kächele [24] | 2014 | Germany | cross-sectional study | NA | Nondrinkers moderate drinker:0–20g/day heavy drinkers: >20g/day | 432 | NA | 265/167 | 142 N:35(100) M:19(100) H::88(315) | Ultrasonography | Interview |
| Liu [25] | 2014 | China | cross-sectional study | NA | Non-drinkers <10g/day | 528 | 53.4±8.2 | W:528 | 121 N:89(398) D:32(130) | Ultrasonography | Questionnaire |
| Lau [26] | 2015 | Germany | cross-sectional study | NA | Non-drinkers .<20g/day 20–40g/day >40 g/day | 1901 | 50.5±20.9 | W:1901 | 413 N:98(322) <20:300(1480) 20–40:14(88) >40:1(11) | Ultrasonography | Standardised interview |
| Hara [27] | 2019 | Japan | cross-sectional study | NA | Non-drinkers Light drinkers: 0–19g/day | 1190 | 55.9±8.9 | M:1190 | 561 N:269(505) L:292(685) | Ultrasonography | Questionnaire |
| Tan [28] | 2020 | Malaysia | cross-sectional study | NA | Non-drinkers Modest: ≤21 units per week in men ≤14 units per week in women | 557 | 61.4±10.8 | 226/177 | 403 N:342(465) M:61(92) | Transient elastography and liver biopsy | Review |
| Peeraphatdit [29] | 2020 | USA | retrospective cohort study | 5.8Y | Non-drinkers Moderate drinkers: ≤2 drinks/day Heavy drinkers: >2 drinks/day | 18506 | 55.8±16.9 | 6703/11803 | 684 N:132(3657) M:505(14236) H:47(613) | Ultrasonography, computed tomography, or magnetic resonance imaging | Questionnaire |
| Chang [30] | 2020 | South Korea | cohort study | 15.7Y | Non-drinker Light drinkers: 1-10g/day Moderate drinkers: 10-20g/day for women and 10-30g/day for men | 190048 | 35.5±6.5 | 83967/106081 | 43466 N:11915(60443) L:16974(84241) M:14577(45364) | Ultrasonography | Questionnaires |

M: Men; W: Woman; N: Non-drinker; L:Light drinker M: Moderate drinker; H: Heavy drinker; Y: Year

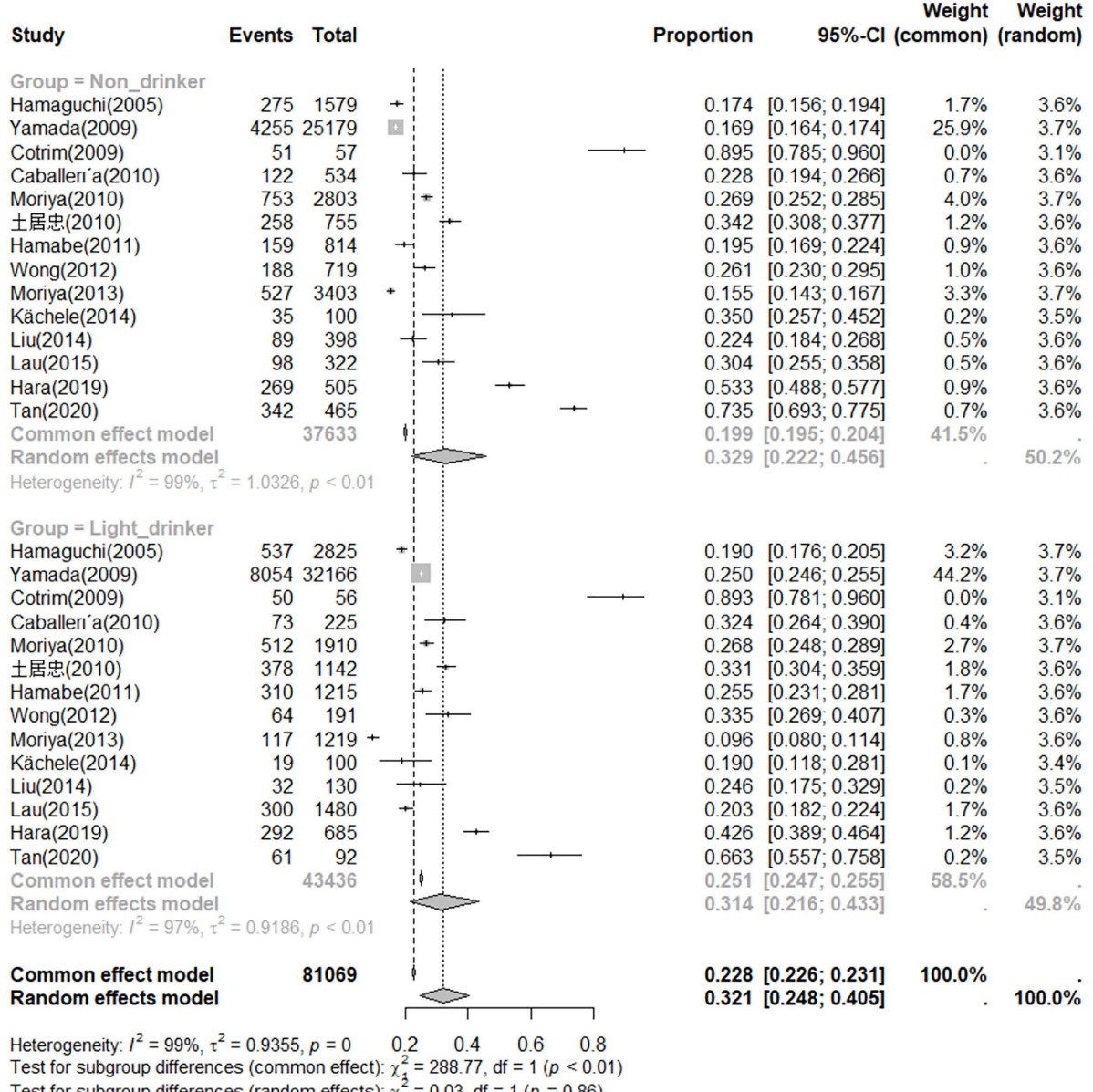

**Fig 2. Prevalence of NAFLD in non-drinkers and light drinkers.**

represented by the other countries, we evaluated the relative risk (RR) of the NAFLD prevalence between light drinkers and non-drinkers. The random-effects model showed no significant difference between these two groups, with an RR of 0.96 (95% CI, 0.76–1.21) (Fig 4).

Subgroup analyses by gender were conducted, with seven studies reporting the prevalence of NAFLD in both men and women. Among these studies, four reported that the prevalence of NAFLD in male light drinkers was lower than that in non-drinkers, indicating that light drinking reduced the prevalence of NAFLD (RR = 0.82, 95% CI, 0.79–0.85) (Fig 5). In contrast, six studies reported the prevalence of NAFLD in female light drinkers compared to non-drinkers, and

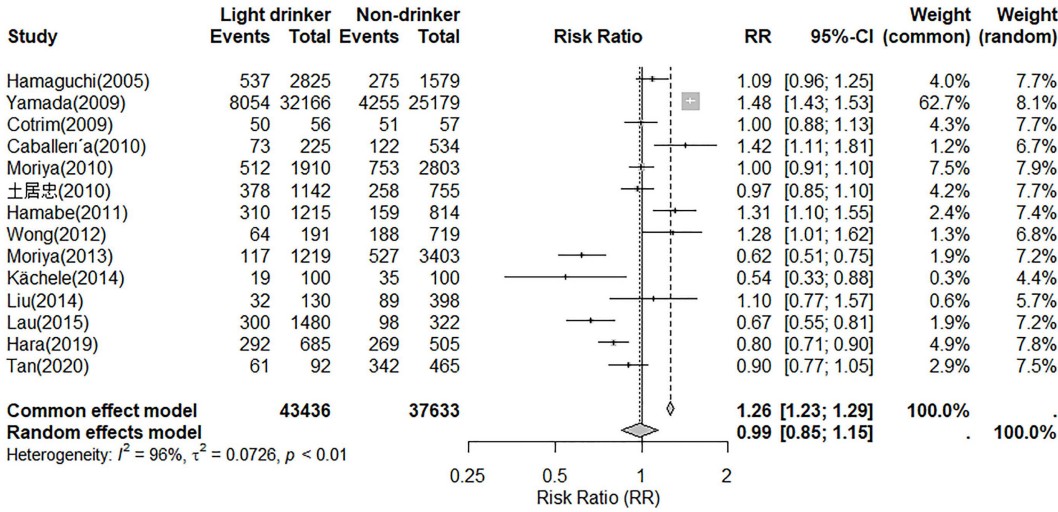

**Fig 3. Relative risk of the relationship between light drinkers and NAFLD compared to non-drinkers.**

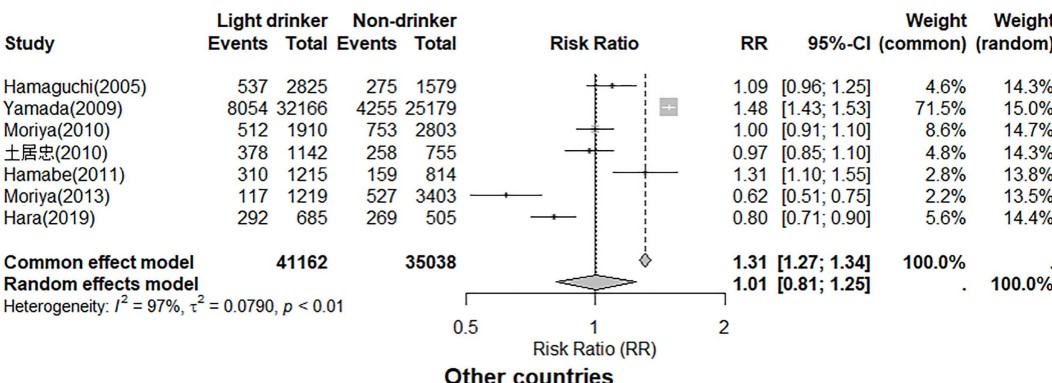

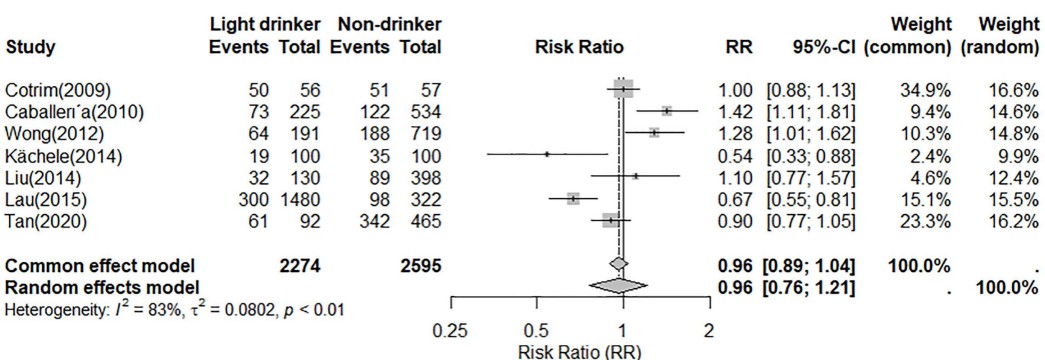

**Fig 4. Relative risk of the relationship between light drinkers and NAFLD compared to non-drinkers in Japan group and the other countries group.**

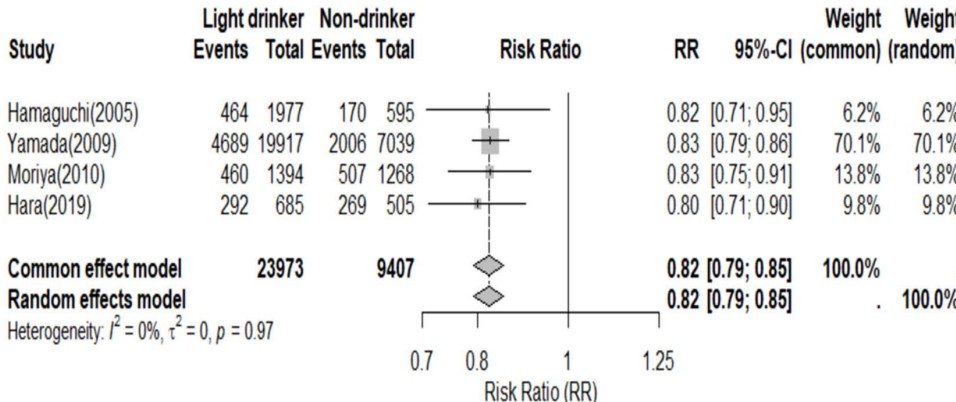

**Fig 5. Prevalence of NAFLD in men light drinkers and non-drinkers.**

the random-effects model showed no significant difference in prevalence between these groups (RR = 0.90, 95% CI, 0.60–1.36) (Fig 6).

**Quality appraisal.** The results of the quality appraisal are presented in the S1 Table in S1 File. All the studies were rated as high quality. Eleven studies either did not include all patients from the specified time period or failed to report whether subjects were consecutive. Additionally, eleven studies did not report any assessments undertaken for quality assurance purposes. One study did not specify the time period used for identifying patients, and one did not report the response rate of patients.

## Incidence

Four studies investigated the incidence of NAFLD, encompassing 68,894 non-drinkers and 149,992 light drinkers. The incidence of NAFLD was found to be 0.0126 cases per person-year among non-drinkers, while it was 0.0153 cases per person-year among light drinkers. The relative risk (RR) of the relationship between light drinkers and NAFLD, compared to non-drinkers, showed an advantage for non-drinkers (RR = 1.18, 95% CI, 1.08–1.30) (Fig 7).

**Quality appraisal.** The results of the quality appraisal are presented in the S2 Table in S1 File. All the studies were rated as high quality. However, all four studies exhibited shortcomings in terms of comparability.

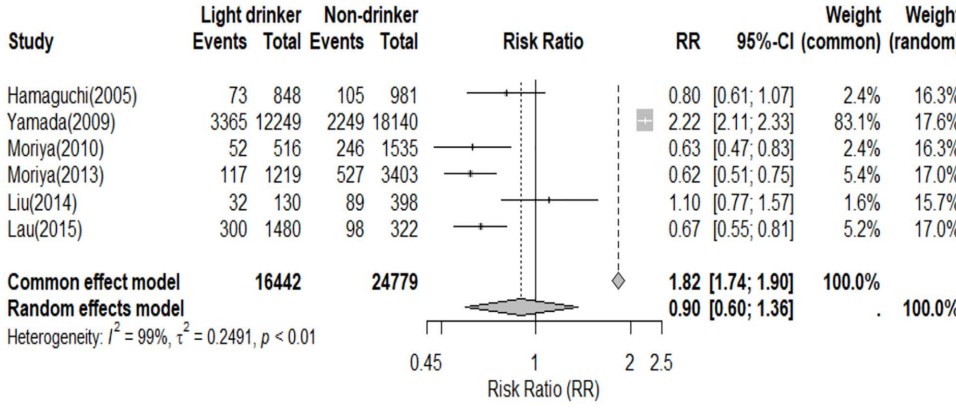

**Fig 6. Prevalence of NAFLD in women light drinkers and non-drinkers.**

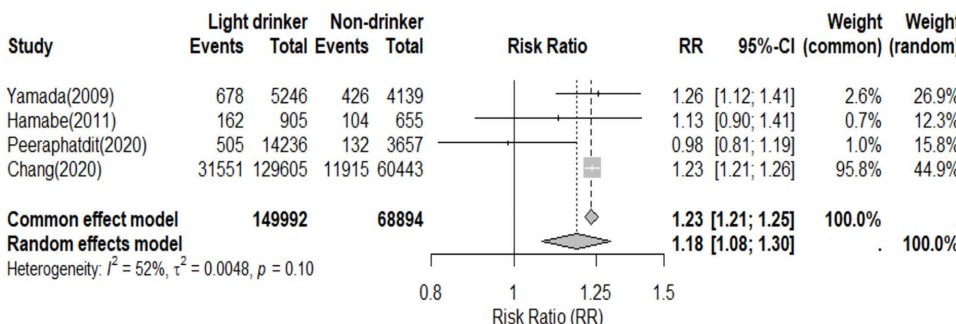

**Fig 7. Incidence of NAFLD in non-drinkers and light drinkers.**

**Publication bias.** There was no evidence of publication bias for any of the estimates of prevalence and incidence. This conclusion was based on visual inspection of funnel plots and the Begg and Egger tests (all P > 0.05) (S1-S4 Figs in S1 File).

## Discussion

This systematic review and meta-analysis compared the effects of light drinking and non-drinking on the prevalence and incidence of NAFLD. Additionally, subgroup analyses were performed to examine the influence of country and gender on prevalence and incidence rate.

Previous systematic reviews on the prevalence and incidence of NAFLD in light drinkers and non-drinkers did not adequately define "light drinking" and excluded many recent clinical studies. Contrary to previous findings, our study shows no significant difference in the prevalence of NAFLD between light drinkers and non-drinkers (RR = 0.99, 95% CI, 0.85–1.15). However, regarding incidence, our results differ substantially from previous studies. We found that the incidence of NAFLD is lower in non-drinkers than in light drinkers (RR = 1.18, 95% CI, 1.08–1.30),which may be linked to alcohol's effect on lipid metabolism [31]. Additionally, unlike previous studies, studies from Japan, China, Europe, and the Americas show no difference in the impact of light drinking versus non-drinking on NAFLD prevalence.

In the gender subgroup analysis, we found that light drinking benefits men, while no significant difference was observed in women. This result aligns with the majority of studies from Japan included in the literature [16,19], which emphasize the benefits of alcohol consumption in NAFLD, with a predominance of positive effects observed in men. In terms of incidence, non-drinkers exhibited an advantage, differing from the results of previous studies.

Considering the heterogeneity among studies, single pooled prevalence and incidence estimates should be interpreted cautiously. To address this heterogeneity, we used a random-effects model for assessments. Although we conducted subgroup analyses to explore several factors that might partially explain the high heterogeneity, including gender and country, it was impossible to account for all these factors in a single model simultaneously.

This study included 299,955 individuals, comprising 106,572 non-drinkers and 193,428 light drinkers, among whom there were 63,693 patients diagnosed with NAFLD. We also conducted subgroup analyses. However, the study is not without limitations. The number of studies comparing incidence rates remains small, leading to some discrepancies in the results. Furthermore, the inclusion of articles was restricted due to our stringent criteria regarding alcohol consumption. And there is heterogeneity among the included studies, possibly due to differences in ethnicity, country, sample size, and diagnostic methods. Because few studies assessing these factors (for example, studies from countries other than Japan are still lacking), conducting meta-analyses for specific subgroups is impossible. Ideally, a multivariable meta-regression would have been employed to address the possible confounding effects of variables such as age and location. However, this would require a large number of studies, so only stratified estimates are provided.

Overall, we found that non-drinking is closely associated with a lower prevalence and incidence of NAFLD compared to light drinking. This finding diverges notably from previous similar meta-analyses. Although this finding aligns with the increasing view that light drinking is a risk factor for NAFLD, it should still be interpreted with caution. This finding was likely due to the significant heterogeneity (>99% for prevalence studies) between these two groups of studies. Nonetheless, this study serves as a reminder of the risks associated with alcohol consumption and may offer valuable insights for the development of future guidelines and policies.

This systematic review and meta-analysis presents information about the impact of light alcohol consumption on NAFLD based on international studies. The number of such studies in other countries is minimal except for Japan. These are considerable gaps that must be addressed in future work. Future research should focus on the potential impact of alcohol consumption assessment on the reporting of NAFLD, as it might explain the significant differences in study results. Methodological factors leading to study heterogeneity, including data collection methods, sources of case identification, and evaluation criteria, should also be explored. Additionally, more clinical studies are needed to compare the impact of light drinking versus non-drinking on the prevalence and incidence of NAFLD with unified protocols and standards. And carefully designed prospective studies and randomized controlled trials on the effect of moderate alcohol use in patients with NAFLD are needed to clarify this controversial question.

## Supporting information

**S1 File.** S1 Appendix. Complete list of search terms. S1 Table. The quality appraisal of prevalence studies. S2 Table. The quality appraisal of incidence studies. S1 Fig. Funnel plot analysis of publication bias for the incidence of NAFLD. S2 Fig. Funnel plot analysis of publication bias in male NAFLD prevalence. S3 Fig. Funnel plot analysis of publication bias in female NAFLD prevalence. S4 Fig. Funnel plot analysis of publication bias for the prevalence of NAFLD. (ZIP)

## Author contributions

**Data curation:** JingKai Yuan, Zhiheng Chen, YuKai Gu, Yaobin Liang, Zhiqi Yao.

**Formal analysis:** JingKai Yuan.

**Investigation:** JingKai Yuan, YuKai Gu, Yaobin Liang.

**Methodology:** JingKai Yuan, Zhiqi Yao.

**Project administration:** JingKai Yuan.

**Supervision:** Zhiqi Yao.

**Validation:** Zhiqi Yao.

**Writing – original draft:** JingKai Yuan.

**Writing – review & editing:** Zhiqi Yao.

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
