## [Decision Letter · Decision Letter 0]

7 Jan 2025

PONE-D-24-50739Effects of alcohol consumption on the prevalence and incidence of Non-alcoholic Fatty Liver Disease: a systematic review and meta-analysisPLOS ONE

Dear Dr. Yao,

Thank you for submitting your manuscript to PLOS ONE. After careful consideration, we feel that it has merit but does not fully meet PLOS ONE’s publication criteria as it currently stands. Therefore, we invite you to submit a revised version of the manuscript that addresses the points raised during the review process.

We look forward to receiving your revised manuscript.

Kind regards,

Anna Di Sessa, PhD, MD

Academic Editor

PLOS ONE

Journal Requirements:

4. As required by our policy on Data Availability, please ensure your manuscript or supplementary information includes the following:

Reviewers' comments:

Reviewer's Responses to Questions

**Comments to the Author**

1. Is the manuscript technically sound, and do the data support the conclusions?

Reviewer #1: Yes

Reviewer #2: Yes

2. Has the statistical analysis been performed appropriately and rigorously? 

Reviewer #1: Yes

Reviewer #2: Yes

3. Have the authors made all data underlying the findings in their manuscript fully available?

Reviewer #1: Yes

Reviewer #2: Yes

4. Is the manuscript presented in an intelligible fashion and written in standard English?

Reviewer #1: Yes

Reviewer #2: Yes

5. Review Comments to the Author

Reviewer #1: REVIEWER SUGGETIONS AND COMMENT

Generally its good idea on differentiating the non-alcoholic and light alcoholic on developing of the NAFLD I was read through out of the document several issues should be improve in order when the reader search kind of this article should it be well understandable the information provided.

Adhere to the journals guideline in organizing the work

Work extensively to be clear grammar and typographical errors throughout the document

ABSTRACT

The authors should update and make it scientific sound when reading part of the background. Additionally, the purpose needs to be revised. Also I noted the authors did not address the conclusion revise and improve

INTRODUCTION

On part of introduction the authors should improve it’s very narrow some of information when reading brings confusion.

METHODS

On part of study design the authors should make it clear. Will make the confusion to the reader.

Reporting bias assessment: The author should mention any procedures used to assess the likelihood of bias in a synthesis due to missing results (caused by reporting biases).Also, identify any methodologies used to determine certainty (or confidence) in the body of evidence supporting an outcome.

DATA COLLECTION PROCESS

The authors should put it clear on specifying the method used to collect the data and the tools used

RESULT

The authors should arrange the section of the result well. Review and improve.

The authors should cite those studies which might appear to meet the inclusion criteria and explain why excluding them.

Risk of bias of each study included

Also the authors should summaries the characteristic and risk of bias among contributing studies

Also present the result of risk of bias to missing results arising from reporting of bias

Present the result of possible cause of heterogeneity among study results

Figure and table authors should improve are not clear and understandable

DISCUSSION

Provide a general interpretation of the results in the context of other evidence.

Discuss limitations of the evidence included in the review.

Discuss limitations of the review processes used.

Discuss implications of the results for practice, policy, and future research.

REFERENCES

Several references do not fit the requirements of Vancouver style. Revise and improve them

Reviewer #2: introduction,

paragraph 2: please superscript ref 5

"However, these studies defined light drinking

as less than 40 g/day10 and 20 g/day11, respectively" what did you mean by respectively? men and women?

method section:

did you assess the quality of studies? please add method of quality appraisal to the method section.

resuts:

page 5: "The 16 studies included in the NAFLD prevalence analysis comprised a sample population of 81,069 individuals" in this sentence 16 should be 14

DISCUSSION:

1. PLEASE discuss the reasons for differences between studies.

2. please references to disussion section. none of the sentences in this section has references

references: all included studies should be referenced.

figure 1:

in reason section, two last lines are repeated.

table 1:

it is better to add a coloumn about the result of each study

it is better to add a coloumn about the factors that use for adjustment in associaion analysis.

6. PLOS authors have the option to publish the peer review history of their article (what does this mean?). If published, this will include your full peer review and any attached files.

Reviewer #1: **Yes: **rehema abdallah

Reviewer #2: No

---

## [Author Response · Author response to Decision Letter 1]

25 Feb 2025

Dear Editors:

Thank you for your valuable feedback and for the opportunity to revise and resubmit our manuscript entitled "Effects of alcohol consumption on the prevalence and incidence of non-alcoholic fatty liver disease: a systematic review and meta-analysis". We greatly appreciate the time and effort you and the reviewers have taken to evaluate our work. We have carefully considered all of the comments and suggestions, and we believe that the revisions have significantly improved the quality and clarity of our manuscript.

Below, we provide a detailed response to each of the reviewers’ comments and outline the changes made:

Journal Requirements:

1.We are sorry for our careless mistakes. And we have revised the manuscript to comply with PLOS ONE's style requirements. However, I regret to note that I inadvertently failed to activate the revision tracking feature during the formatting adjustments, which resulted in the absence of a documented record of the formatting changes made.The data we used was already provided as part of the article, so we believe it is unnecessary to deposit it in the appropriate public repositories.

2.All data that we generated or analyzed in this study are included in the literature included in this article.

3.We had updated our ORCID account.

4.All literature retrieved through the search process, along with the articles screened and ultimately included, were systematically organized in a table titled "Retrieved Literature." The reasons for excluding specific studies are detailed in Fig 1. All included studies have been published, and the data extracted from each study are comprehensively presented in a table within the article. Additionally, I conducted a risk of bias and quality assessment for each included study, as outlined in the manuscript. The corresponding tables are available in the Appendix for reference.

Reviewer #1

Thank you very much for your valuable feedback. We sincerely apologize for the mistakes we made. We have made the necessary changes to the grammar and typography and have ensured compliance with the journal's guidelines.

Abstract:

Thank you for your suggestion. However, the background section is based on epidemiological studies, which I believe makes it scientifically sound. I have also added a conclusion and clarified the purpose more explicitly.

Introduction:

We thank the reviewer for pointing out this issue. And we have followed your suggestions to make several improvements to the introduction section to make it clearer for the reader.

Methods:  

The methodology section has been revised for greater clarity and ease of understanding for the reader. And our study included only articles with complete data, eliminating any instances of incomplete results.

Data collection process:

We thank the reviewer for pointing out this issue and we have specified that all meta-analytic analyses were conducted using the meta packages in R statistical software (version 4.3.1).

Result

The number of excluded articles is high, and the main reasons are listed in the flowchart. The risk of bias is presented in the RESULTS section, with the related icon in the Appendix 1. The relevant data and characteristics of each included study are provided in Table I, and the possible reasons for heterogeneity among the study results are elaborated in the DISCUSSION.

Discussion

We have made changes to the article and provide a general interpretation of the results in the context of other evidence. We also address the limitations of this article and provide a detailed discussion on the potential implications of its findings.

Reviewer #2: 

Thank you for reviewing our article so thoroughly and for your generous comments!

Introduction:

We had superscripted ref 5.

The word "respectively" has been removed, and the sentence has been revised to: “ However, these studies defined light drinking as less than 40 g/day for man and 20 g/day for women”

Method section:

We had assesed the quality of studies and added method of quality appraisal to the method section.

Result:

Thank you very much for pointing out my mistake; I have revised it based on your comments.

Discussion:

After making corrections, we discussed the potential differences and cited relevant references.

References:

Thanks for the advice. All the included studies had been referenced

Figure 1

Thank you very much for pointing out my mistake; I have revised it based on your comments.

Table 1

Thank you for your suggestion. Initially, I did consider including it, but due to the limited space in the table, I decided not to include the results and adjustment factors. Based on considerations of space and practical significance, I felt their inclusion was not essential. However, if the editors believe that including these two results would improve the paper, I am willing to revise it. Once again, thank you very much!

We hope that these revisions address the concerns raised and improve the manuscript. We are grateful for the constructive feedback and believe that it has strengthened our work. Please find the revised manuscript attached for your consideration.

Thank you again for your time and support. We look forward to your response.

Sincerely,

---

## [Decision Letter · Decision Letter 1]

27 Apr 2025

PONE-D-24-50739R1Effects of alcohol consumption on the prevalence and incidence of Non-alcoholic Fatty Liver Disease: a systematic review and meta-analysisPLOS ONE

Dear Dr. Yao,

Thank you for submitting your manuscript to PLOS ONE. After careful consideration, we feel that it has merit but does not fully meet PLOS ONE’s publication criteria as it currently stands. Therefore, we invite you to submit a revised version of the manuscript that addresses the points raised during the review process.

We look forward to receiving your revised manuscript.

Kind regards,

Anna Di Sessa, PhD, MD

Academic Editor

PLOS ONE

Journal Requirements:

Reviewers' comments:

Reviewer's Responses to Questions

**Comments to the Author**

1. If the authors have adequately addressed your comments raised in a previous round of review and you feel that this manuscript is now acceptable for publication, you may indicate that here to bypass the “Comments to the Author” section, enter your conflict of interest statement in the “Confidential to Editor” section, and submit your "Accept" recommendation.

Reviewer #1: All comments have been addressed

Reviewer #2: (No Response)

2. Is the manuscript technically sound, and do the data support the conclusions?

Reviewer #1: Yes

Reviewer #2: Yes

3. Has the statistical analysis been performed appropriately and rigorously? 

Reviewer #1: Yes

Reviewer #2: Yes

4. Have the authors made all data underlying the findings in their manuscript fully available?

Reviewer #1: Yes

Reviewer #2: Yes

5. Is the manuscript presented in an intelligible fashion and written in standard English?

Reviewer #1: Yes

Reviewer #2: Yes

6. Review Comments to the Author

Reviewer #1: Reviewer comment and suggestions

I was able to examine closely and realize that the author has been able to give paths for each step of writing the procedures in manuscript; however there are a few things authors need to include and make it clear. Use Level 1 heading for all major sections (Abstract, Introduction, Materials and methods, Results, Discussion, etc.).

• Adhere to the journals guideline in organizing the work especially PLOS ONE

ABSTRACT the authors should make the conclusions according to result.

RESULT the authors should arrange the section of the result well. Review and improve.

Thank you.

Reviewer #2: Dear editor

Although many issues raised by reviewers are responsed; i think that as the included studies are analytical, it is important to add the resukt and adjustment variables for each study in table q; or oresent these data as a supplementary table.

7. PLOS authors have the option to publish the peer review history of their article (what does this mean?). If published, this will include your full peer review and any attached files.

Reviewer #1: **Yes: **rehema abdallah

Reviewer #2: No

---

## [Author Response · Author response to Decision Letter 2]

12 Jun 2025

We have made the changes and appreciate for the suggestions!

---

## [Decision Letter · Decision Letter 2]

28 Jul 2025

Effects of alcohol consumption on the prevalence and incidence of Non-alcoholic Fatty Liver Disease: a systematic review and meta-analysis

PONE-D-24-50739R2

Dear Dr. Yao,

We’re pleased to inform you that your manuscript has been judged scientifically suitable for publication and will be formally accepted for publication once it meets all outstanding technical requirements.

Kind regards,

Anna Di Sessa, PhD, MD

Academic Editor

PLOS ONE

Additional Editor Comments (optional):

Reviewers' comments:

Reviewer's Responses to Questions

**Comments to the Author**

1. If the authors have adequately addressed your comments raised in a previous round of review and you feel that this manuscript is now acceptable for publication, you may indicate that here to bypass the “Comments to the Author” section, enter your conflict of interest statement in the “Confidential to Editor” section, and submit your "Accept" recommendation.

Reviewer #2: All comments have been addressed

2. Is the manuscript technically sound, and do the data support the conclusions?

Reviewer #2: Yes

3. Has the statistical analysis been performed appropriately and rigorously? 

Reviewer #2: Yes

4. Have the authors made all data underlying the findings in their manuscript fully available?

Reviewer #2: Yes

5. Is the manuscript presented in an intelligible fashion and written in standard English?

Reviewer #2: Yes

6. Review Comments to the Author

Reviewer #2: this systematic review is revised according to comments. It is now acceptable in its present form .

7. PLOS authors have the option to publish the peer review history of their article (what does this mean?). If published, this will include your full peer review and any attached files.

Reviewer #2: No

---

## [Editor Report · Acceptance letter]

PONE-D-24-50739R2

PLOS ONE

Dear Dr. Yao,

I'm pleased to inform you that your manuscript has been deemed suitable for publication in PLOS ONE. Congratulations! Your manuscript is now being handed over to our production team.

Kind regards,

on behalf of

Dr. Anna Di Sessa

Academic Editor

PLOS ONE